# Altered Expression of Intestinal Tight Junction Proteins in Heart Failure Patients with Reduced or Preserved Ejection Fraction: A Pathogenetic Mechanism of Intestinal Hyperpermeability

**DOI:** 10.3390/biomedicines12010160

**Published:** 2024-01-12

**Authors:** Eleni-Evangelia Koufou, Stelios F. Assimakopoulos, Pinelopi Bosgana, Anne-Lise de Lastic, Ioanna-Maria Grypari, Georgia-Andriana Georgopoulou, Stefania Antonopoulou, Athanasia Mouzaki, Helen P. Kourea, Konstantinos Thomopoulos, Periklis Davlouros

**Affiliations:** 1Department of Cardiology, Patras University Hospital, 26504 Patras, Greece; elenikoufou17@gmail.com; 2Department of Internal Medicine and Division of Infectious Diseases, University of Patras Medical School, 26504 Patras, Greece; sassim@upatras.gr; 3Department of Pathology, Medical School of Patras, 26504 Patras, Greece; bosgana.p@gmail.com (P.B.); hkourea@yahoo.com (H.P.K.); 4Laboratory of Immunohematology, Division of Hematology, Department of Internal Medicine, Medical School, University of Patras, 26504 Patras, Greece; delastic@gmail.com (A.-L.d.L.); mouzaki@upatras.gr (A.M.); 5Cytology Department, Aretaieion University Hospital, National Kapodistrian University of Athens, 11528 Athens, Greece; iomagry@yahoo.gr; 6Department of Nephrology and Transplantation, Patras University Hospital, 26504 Patras, Greece; georgop91@gmail.com; 7Department of Medicine, University of Patras, 26504 Patras, Greece; stephanie.n.ant@gmail.com; 8Division of Gastroenterology, Department of Internal Medicine, Medical School, University of Patras, University Hospital of Patras, 26504 Patras, Greece; kxthomo@hotmail.com

**Keywords:** heart failure, intestinal hyperpermeability, systemic endotoxemia, systemic inflammation, tight junction dysfunction

## Abstract

Although intestinal microbiota alterations (dysbiosis) have been described in heart failure (HF) patients, the possible mechanisms of intestinal barrier dysfunction leading to endotoxemia and systemic inflammation are not fully understood. In this study, we investigated the expression of the intestinal tight junction (TJ) proteins occludin and claudin-1 in patients with HF with reduced (HFrEF) or preserved ejection fraction (HFpEF) and their possible association with systemic endotoxemia and inflammation. Ten healthy controls and twenty-eight patients with HF (HFrEF (n = 14), HFpEF (n = 14)) underwent duodenal biopsy. Histological parameters were recorded, intraepithelial CD3+ T-cells and the expression of occludin and claudin-1 in enterocytes were examined using immunohistochemistry, circulating endotoxin concentrations were determined using ELISA, and concentrations of cytokines were determined using flow cytometry. Patients with HFrEF or HFpEF had significantly higher serum endotoxin concentrations (*p* < 0.001), a significantly decreased intestinal occludin and claudin-1 expression (in HfrEF *p* < 0.01 for occludin, *p* < 0.05 for claudin-1, in HfpEF *p* < 0.01 occludin and claudin-1), and significantly increased serum concentrations of IL-6, IL-8, and IL-10 (for IL-6 and IL-10, *p* < 0.05 for HFrEF and *p* < 0.001 for HFpEF; and for IL-8, *p* < 0.05 for both groups) compared to controls. Occludin and claudin-1 expression inversely correlated with systemic endotoxemia (*p* < 0.05 and *p* < 0.01, respectively). Heart failure, regardless of the type of ejection fraction, results in a significant decrease in enterocytic occludin and claudin-1 expression, which may represent an important cellular mechanism for the intestinal barrier dysfunction causing systemic endotoxemia and inflammatory response.

## 1. Introduction

According to the current pathogenetic aspects of chronic heart failure (HF), systemic inflammation plays a central role in the progression of the disease [1]. HF with reduced ejection fraction (HFrEF) leads to intestinal congestion, which subsequently impairs the integrity and function of the intestinal epithelial barrier and increases intestinal permeability [2]. In addition, previous studies have shown that the composition of the intestinal microbiota is altered (dysbiosis), which contributes to intestinal barrier dysfunction [3,4,5]. Increased intestinal permeability promotes the translocation of bacteria and endotoxins into the systemic circulation, which then triggers a systemic inflammatory response (the “gut hypothesis” of inflammation in HF) [4,5]. The possible mechanisms of increased intestinal permeability in HF patients have not been adequately investigated, especially in patients with preserved ejection fraction (HFpEF), although systemic inflammation has also been demonstrated [6,7].

The major regulator of paracellular permeability and the natural barrier against the influx of harmful contents from the gut lumen into the internal milieu is the apical junctional complex, which consists of the tight junctions (TJs), the underlying adherens components, and desmosomes [8,9,10]. The TJs are composed of various transmembrane and cytosolic proteins, including occludin, members of the claudin family, junctional adhesion molecules, cingulin, and tricellulin, which are linked to the actin cytoskeleton via the intracellular proteins of the zonula occludens family [11]. Epithelial TJs represent an incredibly dynamic structure that open or close the paracellular route in response to a variety of stimuli, with occludin and claudins playing a central role in this regulation [12,13].

The aim of this study was to investigate the possible presence of alterations in the intestinal epithelial barrier in patients with HFpEF compared with HFrEF, with a particular focus on the expression of the TJ proteins occludin and claudin-1, and to explore a possible relationship with the development of systemic endotoxemia and inflammation.

## 2. Materials and Methods

### 2.1. Study Design

The study was approved by the Ethics Committee of the Hospital (N.61/12.12.2018), in accordance with the Declaration of Helsinki as revised in 2000. Written informed consent was obtained from all participants in the study. This was a pilot prospective, nonrandomized study of 10 healthy control participants (group A) and 28 patients with HF in stable condition, either HFrEF (group B, n = 14) or HFpEF (group C, n = 14), who underwent duodenal biopsy. All biopsies were obtained from the second part of the duodenum, distal to the ampulla of Vater. Inclusion criteria for the study were age of >18 years and a diagnosis of HF (on clinical grounds and by echocardiography and biomarkers). Exclusion criteria were age of <18 years, an unstable condition of HF (recent hospitalization for HF or recent change in chronic HF medication within 4 weeks), severe valvular heart disease, significant pulmonary disease on pulmonary function tests or pulmonary imaging, chronic pulmonary embolism, hypertrophic cardiomyopathy or previous heart transplantation, a history of malignancy, rheumatologic disease, severe chronic kidney disease (stage 4 or 5 according to the National Kidney Foundation), diabetes mellitus type 1 or 2, obesity (body mass index ≥ 30), gastrointestinal disorders (celiac disease, inflammatory bowel disease, irritable bowel syndrome, or gastrointestinal bleeding within 4 weeks), intestinal surgery, active inflammation, alcohol abuse or use of antibiotics, corticosteroids, nonsteroidal anti-inflammatory drugs, and antioxidant substances (vitamin C or E, allopurinol, or N-acetyl-cysteine) within 1 month. Diabetes and obesity are commonly associated with HFpEF but were excluded because it has been previously demonstrated that these comorbidities are associated with the disruption of gut barrier function and enterocytes’ TJs integrity [14,15,16]. The same applies for the other medical conditions included in the exclusion criteria of the study. The control group consisted of people without any of the above exclusion criteria or any known comorbidity, who underwent an upper gastrointestinal tract endoscopy for dyspeptic symptoms, after consultation with a gastroenterologist, without any pathological findings on endoscopic examination.

### 2.2. Endotoxin and Cytokines Measurements

Before endoscopy, blood was drawn from a peripheral vein to measure endotoxin and cytokines in all participants. Circulating endotoxin concentration in serum samples was determined via ELISA using commercially available kits, as per the manufacturer’s instructions (Εndotoxin cat#abx051541; Abbexa Ltd., Cambridge Science Park, Cambridge, UK, range = 0.015–2 EU/mL, sensitivity < 0.005), and systemic inflammatory response was assessed by determining cytokine levels (IL-1β, IL-6, IL-8, IL-10, IL-12p70, and TNF-α) using flow cytometry. Cytokine concentration in serum samples was measured with a BD FACS Array Bioanalyzer, using a cytometric bead array (CBA) assay (Human Inflammatory Cytokines Kit, cat#551811, BD Biosciences) (range 20–5000 pg/mL, sensitivity 7.2 pg/mL for IL-1β, 2.5 pg/mL for IL-6, 3.6 pg/mL for IL-8, 3.3 pg/mL for IL-10, 1.9 pg/mL for IL-12, and 3.7 pg/mL for TNF-α). The cytokine concentrations were measured using a Cytometric Bead Array methodology that combines the principles of ELISA and flow cytometry to quantify soluble proteins. As such, protein concentrations that fall below the standard curve of the assay cannot be safely quantified. The samples that fell below range were measured without dilution to ensure the accuracy of the results and subsequently reported as “0”.

### 2.3. Histopathological Evaluation and Immunohistochemistry

All biopsies were fixed in formalin and embedded in paraffin. They were placed in formalin within 10 min of resection and processed within 48 h. Various histologic features were assessed and recorded on each slide stained with hematoxylin and eosin (H&E). These features included architectural villus distortion, epithelial injury on the surface and in the crypts, villous blunting, intraepithelial lymphocytic infiltration on the surface epithelium and in the crypts, the presence and type of cellular infiltration of the lamina propria, the fibrosis of the lamina propria, and the formation of granulation tissue. Apoptotic bodies are defined as round vacuoles containing fragments of karyorrhectic nuclear debris distinct from small, isolated fragments of nuclear chromatin and intraepithelial neutrophils or lymphocytes. Apoptotic bodies and mitoses were counted in all architecturally well-oriented consecutive crypts of the sample, regardless of crypt orientation, and their number per 100 intestinal epithelial cells is referred to as the apoptotic body count and the mitotic count, respectively. For villus length (mm), at least 10 well-oriented villi were evaluated in each sample.

For immunohistochemistry (IHC), serial 3 μm tissue sections were cut, fixed on poly-L-lysine-coated slides, and further processed. The sections were first dried at 25 °C for 1 h, deparaffinized in xylene, and hydrated in gradient alcohol. The antigen was retrieved in Tris/EDTA buffer (pH 9) for 12 min using a pressure antigen retrieval method. Endogenous peroxidase activity was then blocked by incubating the sections with an endogenous peroxidase blocking solution (0.3% H_2_O_2_) for 10 min at room temperature. The sections were then incubated with the following primary antibodies: Claudin-1 (rabbit polyclonal antibody, 1:100, WA314099, cat# 51-9000, Invitrogen, Carlsbad, CA, USA), Occludin (rabbit polyclonal antibody, 1:80, VL314100, cat#71-1500, Invitrogen), and CD3 (rabbit polyclonal antibody, 1:300, Dako, Santa Clara, CA, USA). Dako EnVision polymer (Dako EnVision Mini Flex, Dako Omnis, Agilent Technology Inc., Santa Clara, CA, USA) was used for signal detection. Diaminobenzidine (Dako Omnis, GV823) served as a chromogen, and Harris hematoxylin was used for nuclear counterstaining. Positive and negative controls for antibody validation were used according to the manufacturer’s instructions.

The immunohistochemical expression of occludin, claudin-1, and CD3 was recorded as present (+) or absent (−). For occludin expression, ten well-oriented villi were randomly selected per case, and the percentage of occludin (+) enterocytes was determined by dividing the number of positively stained cells by the total number of enterocytes lining the villi. The localization of staining (nuclear, membranous, or cytoplasmic) was also assessed. In addition, the number of CD3 (+) intraepithelial lymphocytes per 100 intestinal epithelial cells in ten well-oriented villi per case was recorded. Claudin-1 expression was estimated in the intestinal crypts and Brunner glands and expressed as a percentage value of claudin-1 (+) enterocytes by dividing the number of positively stained cells by the total number of intestinal epithelial cells. Photomicrographs were taken with cellSens Entry by Olympus on an Olympus BX41 microscope (Olympus Europa SE & Co., Hamburg, Germany). Three expert pathologists (PB, IMG, and ΕΚ), blinded to the pathologic and clinical features of all cases, performed the histopathologic and immunohistochemical analyses, and when the observers disagreed in their evaluation, consensus was reached by conference on a multi-headed microscope.

### 2.4. Statistical Analysis

Data were analyzed with the statistical package SPSS for Windows (version 25.0; IBM, Armonk, NY, USA) and GraphPad Prism (version 9.1.0, GraphPad Software Inc., San Diego, CA, USA). The normality of the data was tested using the Shapiro–Wilk test, and all parameters except villus length showed a non-normal distribution. Comparisons were made using nonparametric analysis of variance (Kruskal–Wallis test) followed by post hoc Mann–Whitney U test with Bonferroni correction (non-normally distributed data), or using one-way analysis of variance (ANOVA) followed by post hoc Tukey test (normally distributed data). Results are expressed as median (interquartile range) for non-normally distributed data or as mean ± standard deviation for normally distributed data. For the comparison of proportional data, the chi-square test with Yates’ correction was used when necessary. Correlations were estimated with a nonparametric Spearman correlation test. All tests were considered significant if they yielded two-tailed results and a *p*-value of less than 0.05.

## 3. Results

### 3.1. Characteristics of the Patients

The characteristics of the study participants are shown in Table 1. 

Except for age, where controls were younger, no significant differences in gender, incidence of overweight (BMI 25–29.9), or smoking were found between the groups (controls, HFrEF, and HFpEF). Also, there were no differences between HFrEF and HFpEF patients in the presence of hypercholesterolemia, hypertension, or coronary artery disease.

### 3.2. Histopathological Evaluation

No statistically significant differences were found between the control group and the two groups with HF in terms of proliferation (mitotic count), apoptosis, the presence of intraepithelial lymphocytes, or villus length (Table 2).

The statistics are the result of Kruskal–Wallis test for all parameters, except villus length where one-way ANOVA was applied.

### 3.3. Immunohistochemical Results for TJ Proteins and Intraepithelial CD3+ T Cells

In the healthy controls, occludin was expressed as membranous immunostaining mainly in the apical part of the epithelial cells, whereas granular cytoplasmic and subnuclear distribution was also observed. In the healthy controls, almost all epithelial cells lining the villi and the epithelial cells of the crypts showed positive immunostaining for occludin (Figure 1A). 

In patients in both groups B and C, the expression of occludin was greatly reduced in numerous epithelial cells lining the villi (Figure 1B,C), compared with controls (*p* < 0.01 for both groups compared with controls) (Table 3 and Figure 2). 

Interestingly, in both groups B and C, we observed a gradient of occludin immunostaining positivity along the length of the villi, from crypt to tip; occludin expression was maintained in the crypts and basal portion of the villi, and was reduced in the middle part of the villi, whereas a greater loss of expression was observed at the tip of the villi (Table 3). This reduction in occludin expression at the tip of the villi was significant (*p* < 0.05 and *p* < 0.01, for groups B and C, respectively, compared to controls). The reduction in occludin expression in the middle part of the villi was significant only in group C (*p* < 0.01), whereas group B patients showed no statistically significant difference compared to controls.

In healthy subjects, claudin-1 expression was mainly found in Brunner glands and crypts, whereas it was absent in intestinal villi. Claudin-1 expression was significantly decreased in groups B and C compared with controls (*p* < 0.05 and *p* < 0.01, respectively, compared to controls) (Figure 3).

There was no significant difference in intraepithelial CD3+ T cells between groups (Table 2).

### 3.4. Endotoxin and Cytokine Levels

Patients in both groups B and C had significantly higher serum endotoxin concentrations compared with healthy controls (*p* < 0.001) (Figure 4). 

Serum concentrations IL-6 and IL-10 were significantly increased in patients of group B (*p* < 0.05) and group C (*p* < 0.001) compared to controls, as was IL-8 (*p* < 0.05 for both groups compared to controls) (Table 4). 

No statistically significant differences were observed for IL-1β, IL-12p70, or TNF-α (Table 4).

### 3.5. Correlations

In the entirety of patients with HF (pooled patients with HFrEF and HFpEF), the expression of occludin and claudin-1 in the intestinal mucosa (immunohistochemical semi-quantification) was significantly inversely correlated with endotoxin concentration (r = −0.376, *p* < 0.05 for occludin; and r = −0.446, *p* < 0.01 for claudin-1) (Figure 5). 

## 4. Discussion

Recently, there has been increased interest in the theory that gut barrier disruption and increased gut permeability lead to microbial and endotoxin translocation, thereby promoting low-grade inflammation in HF patients, which plays a central role in disease pathogenesis and progression [17,18]. The mechanisms underlying the link between the heart and the gastrointestinal system have not been fully elucidated, but previous studies have shown that a bidirectional relationship exists, leading to the identification and characterization of the so-called “gut–heart axis” [19]. The main pathophysiological mechanism behind the “leaky gut hypothesis” states that the decreased cardiac output observed in patients with HFrEF leads to gut hypoperfusion and mucosal ischemia, compromising the integrity of the gut epithelial barrier and increasing gut permeability [20]. Previous studies have also demonstrated alterations in the composition and function of the gut microbiome, termed dysbiosis, which further contribute to gut barrier dysfunction, as the intestinal microbiota are in constant communication with the intestinal epithelium and the intestinal immune system through the secretion of metabolites and inflammatory mediators [3,4,5]. Through this disrupted gut barrier, gut-derived microbes and endotoxins enter the systemic circulation and trigger a systemic inflammatory response [20]. Therefore, HF leads to a chronic inflammatory state linked to the intestinal barrier, which, together with the neurohormonal axis, leads to disease progression [21].

Although many efforts aimed to find more effective strategies to prevent and modify the course of the disease, the results are still not satisfactory [19]. The presence of different mechanisms that may lead to chronic inflammation and their different pathophysiological pathways may explain why attempts to act on each one have been unsuccessful, and gaps in our knowledge are still evident [21,22]. The present study demonstrates the development of significant cellular alterations in the intestinal mucosa of HFrEF and HFpEF patients that could explain, at least in part, the increased intestinal permeability and endotoxemia. In particular, a significant decrease in occludin and claudin expression in intestinal epithelial cells was observed in both groups of patients compared with controls. The regional loss of occludin expression has already been demonstrated in HFrEF [23]. However, to the best of our knowledge, this is the first study showing that the expression of the TJ proteins occludin and claudin is similarly impaired in the intestinal epithelium of patients with HFpEF. One possible factor mediating the observed changes in intestinal TJs may be systemic endotoxemia itself. Endotoxin can trigger a systemic inflammatory response characterized by the release of cytokines and other proinflammatory mediators that can negatively affect the structure and function of TJs, further promoting the escape of endotoxin from the gut lumen and creating a vicious cycle [24,25]. For occludin in particular, proinflammatory mediators have been shown to downregulate its human promoter, such that higher concentrations of proinflammatory factors in the circulation may lead to decreased occludin expression [25]. Another possible mechanism could be increased oxidative stress. Previous studies have shown that a dysbiotic gut microbial community could contribute to gut barrier dysfunction via the chronic activation of oxidative stress pathways, as higher concentrations of oxidative metabolites in the bloodstream could disrupt the TJ structural complex by modulating the assembly, localization, expression, and function of its components [18,20,24,26,27]. Interestingly, regarding occludin, a specific pattern of expression was observed in the intestinal epithelium in patients with HF regardless of ejection fraction (EF), with a gradually increasing loss of expression from the crypt to the tip of the villi. The same pattern of loss of occludin expression in the intestinal epithelium has been previously described by our group in other diseases associated with increased gut permeability, such as patients with obstructive jaundice or liver cirrhosis [28,29,30,31]. This could be explained by the fact that the microstructure of the intestinal villus surface allows a shunt of oxygenated blood through the base of the villi, exposing the tip of the villi to the risk of relative ischemia. The gradual decrease in tissue oxygen concentration from the villus base to the villus tip is inversely related to blood flow and is directly affected by changes in blood flow [19]. Ischemia is an important factor promoting the destruction of the epithelium of TJs [32].

In the present study, the decreased occludin and claudin-1 expression in the intestinal mucosa was associated with systemic endotoxemia. Our study showed an inverse correlation between intestinal occludin and claudin-1 expression and the extent of systemic endotoxemia. This is consistent with previous studies showing that increased intestinal permeability directly correlates with endotoxemia [17]. The downregulation of enterocytic TJs has been associated with the failure of gut barrier function and increased intestinal permeability in various pathological conditions [23,28,29,30,31,33]. Systemic endotoxemia triggers a systemic inflammatory response, which in turn has deleterious effects on the heart. In the present study, increased levels of cytokines IL-6, IL-8, and IL-10 were found in the systemic circulation of HF patients compared to healthy controls for both groups of HF. It has been previously shown that in HF patients, systemic inflammation leads to oxidative stress and endothelial dysfunction in the myocardium, which can be detected mainly in the coronary microcirculation, followed by a consequent reduction in capillary density that can impair coronary flow reserve and cause diastolic dysfunction [6,7]. This may partly represent the link between gut barrier dysfunction leading to endotoxemia, systemic inflammation, and HF progression. In addition, the gut microbiota-dependent metabolite trimethylamine N-oxide (TMAO) appears to play an important role in maintaining the inflammatory mechanism in HF patients [20,22]. There is evidence in the literature that the dysbiosis of the gut microbiota is associated with high circulating TMAO concentrations due to a disrupted gut mucosal barrier and increased intestinal permeability [22,34]. TMAO activates signaling pathways such as TGF-β1/Smad3 and p65 NF-κΒ, leading to a decrease in energy metabolism and mitochondrial function, and impairs the tricarboxylic acid (TCA) cycle, ultimately adversely affecting myocardial contractile function and intracellular calcium processing, and consequently triggering cardiac hypertrophy and myocardial fibrosis [1,20,22].

There is currently a growing research interest in the therapeutic potential of gut microbiome modulation on gut barrier preservation in diverse pathological entities. Normal gut microbiota through the production of short-chain fatty acids represents an important source of energy for intestinal epithelial cells, while the microbiota-derived pathogen-associated molecular patterns (PAMPs) are recognized by pattern recognition receptors (PRRs) expressed by intestinal immune cells, indicating a relentless reciprocal dialogue between the microbiota and the intestinal immune system. Probiotics are living non-pathogenic microorganisms, which when administered in optimum amounts promote a healthy gut microbiome with health benefits; prebiotics are specific plant fibers that promote the growth of beneficial bacteria; and synbiotics are a combination of the two [35]. Another option for modulating gut microbiota disturbances is the administration of non-absorbable antibiotics [36]. Rifaximin in a non-absorbed, oral administrated antibiotic which has shown beneficial effects on the prevention of bacterial and endotoxin translocation and systemic inflammation in other conditions of gut barrier dysfunction, such as in cirrhotic patients [36]. The role of probiotics, prebiotics, and synbiotics and their beneficial cardiac effects in HF were shown in a small double-blind, placebo-controlled pilot study in 20 HF patients which showed that a probiotic preparation with *Saccharomyces* induced an improvement in cardiac systolic function compared with the placebo group [37]. However, the recently published GutHeart study, which was designed to examine the effect of microbiota modulation on cardiac function in HF patients, showed that neither the probiotic *Saccharomyces boulardii* nor the antibiotic rifaximin affected cardiac function or TMAO [38]. These findings do not preclude the potential role of these treatments in HF patients but rather indicate that a more precise tailored therapeutic approach targeting specific bacterial taxa or a gut-derived metabolite should be attempted.

This study has certain limitations. First, it is a single-center study with a small number of patients. Second, intestinal permeability in the patients of HF was not studied directly (e.g., by a dual sugar absorption test) but indirectly using endotoxemia as a marker. Third, the mechanisms of the disruption of intestinal TJs were not measured, but only attempts were made to describe possible influencing factors such as changes in cytokine concentrations. Fourth, natriuretic peptide levels were not measured, so the stable condition of the heart failure was estimated only with clinical and echocardiographic criteria. Despite the existing limitations, our study demonstrates that altered enterocytic TJ protein expression plays an important role in the pathogenesis of systemic endotoxemia and inflammation in patients with HFpEF in a manner consistent with HFrEF.

## 5. Conclusions

The present study demonstrates for the first time that HF, regardless of its category, induces significant changes in enterocyte TJs, which may represent an important mechanism for gut barrier dysfunction and hyperpermeability at the cellular level. Future studies focusing on the pharmacological modulation of intestinal TJs could lead to a better control of intestinal hyperpermeability in these patients, resulting in fewer complications and better clinical outcomes. The results of the present study, combined with the lack of an effective treatment for HFpEF, underscore the need to find treatments that protect the intestinal barrier and reduce edema, which appear to maintain the pathophysiological mechanisms of HFpEF. The avoidance of enterotoxic agents and the use of personalized interventions to prevent or restore intestinal dysbiosis might have a positive impact in this direction.

## Figures and Tables

**Figure 1 biomedicines-12-00160-f001:**
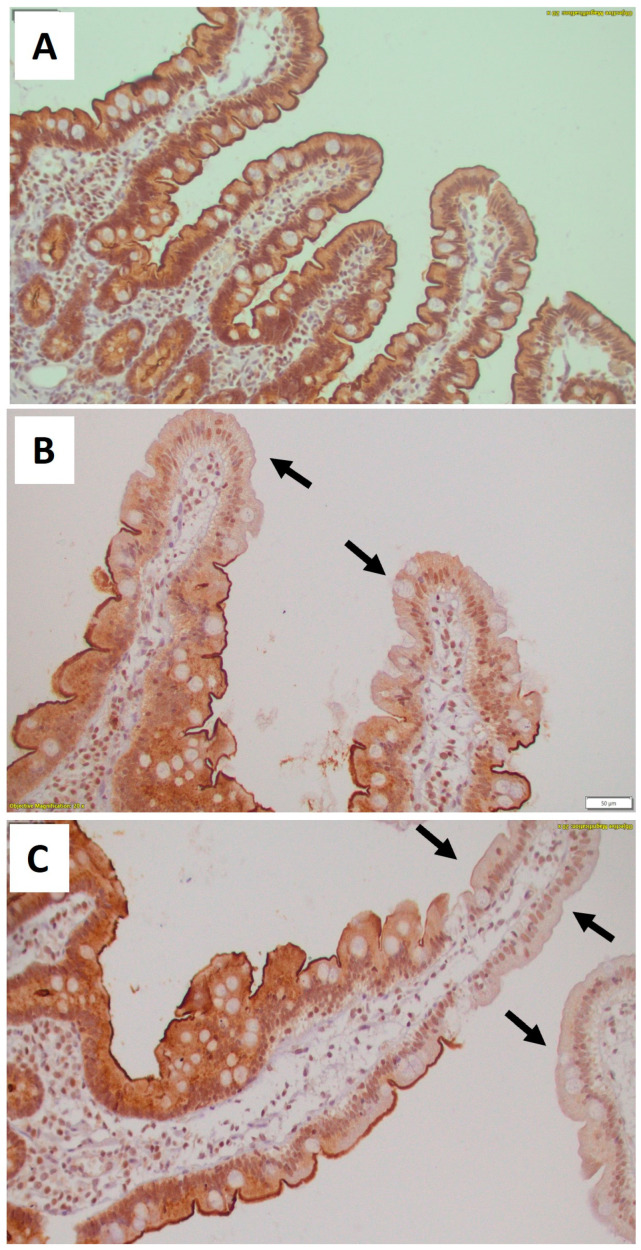
Occludin immunohistochemical expression in the duodenal mucosa. Representative photomicrographs of occludin immunohistochemical expression in the duodenal mucosa. In controls (**A**), almost all epithelial cells lining villi and crypts exhibit positive cytoplasmic and membranous immunostaining for occluding, while there is a pronounced depletion of occludin cytoplasmic and apical membranous expression in numerous epithelial cells mainly at the upper part of the villi (black arrows) in patients with HFrEF (**B**) and HFpEF (**C**). (all microphotographs ×200). HFrEF = heart failure with reduced ejection fraction; HFpEF = heart failure with preserved ejection fraction.

**Figure 2 biomedicines-12-00160-f002:**
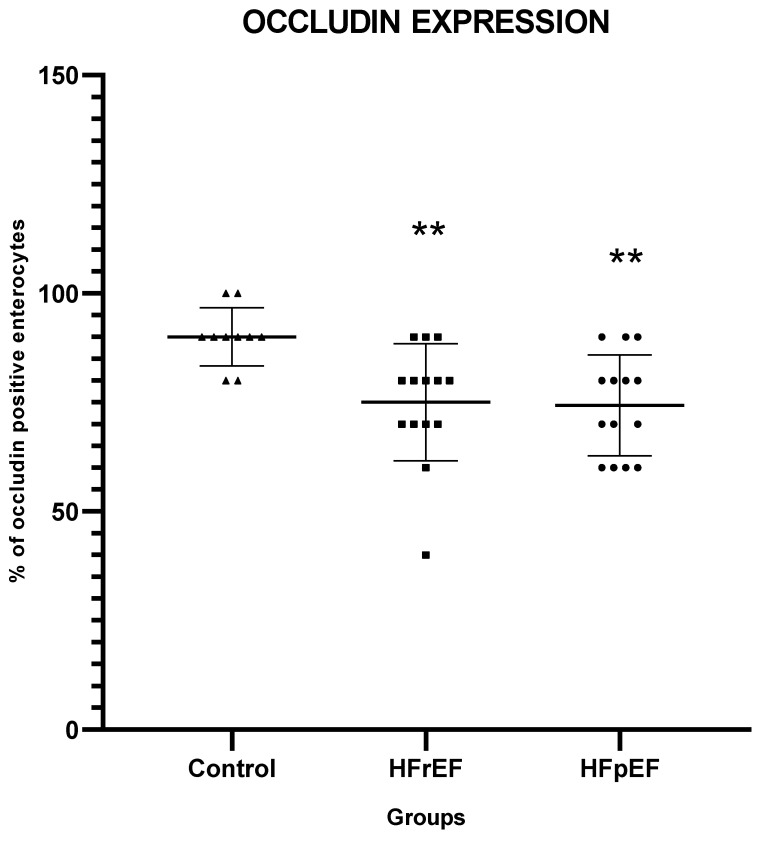
Occludin expression. Occludin expression (% of occludin (+) enterocytes) in the intestinal mucosa of controls and heart failure patients. HFrEF: heart failure reduced ejection fraction; HFpEF: heart failure preserved ejection fraction. ** *p* < 0.01, vs. controls.

**Figure 3 biomedicines-12-00160-f003:**
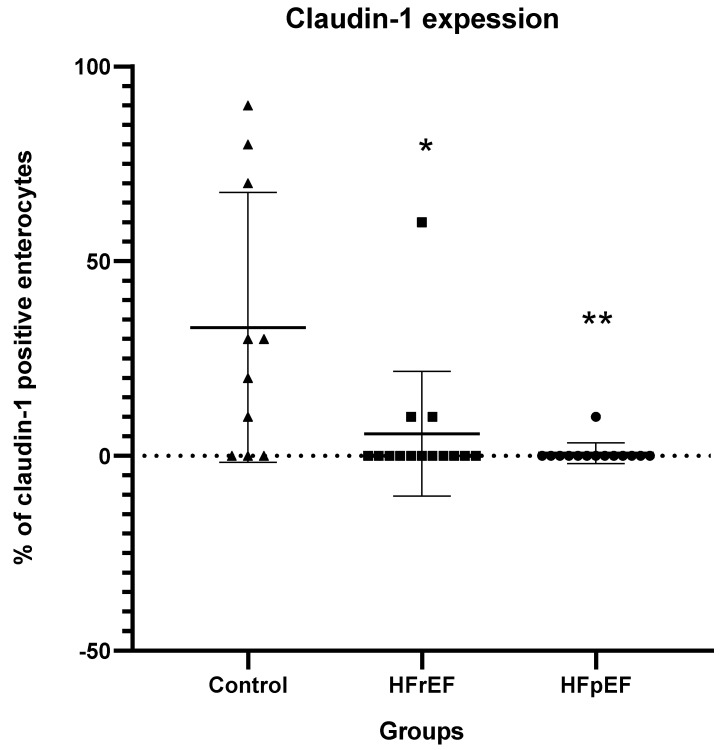
Claudin-1 expression. Claudin-1 expression (% of claudin–1 (+) enterocytes) in the intestinal mucosa. HFrEF: heart failure reduced ejection fraction; HFpEF: heart failure preserved ejection fraction. * *p* < 0.05, ** *p* < 0.01 vs. controls.

**Figure 4 biomedicines-12-00160-f004:**
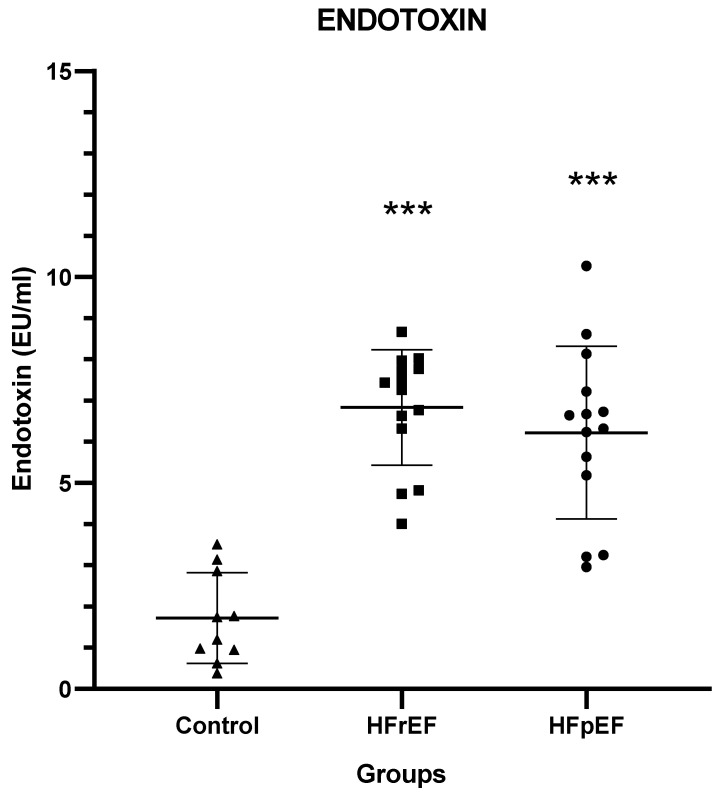
Serum endotoxin concentrations. Εndotoxin concentrations in the peripheral blood of heart failure patients and controls. HFrEF: heart failure reduced ejection fraction; HFpEF: heart failure preserved ejection fraction. *** *p* < 0.001 vs. controls.

**Figure 5 biomedicines-12-00160-f005:**
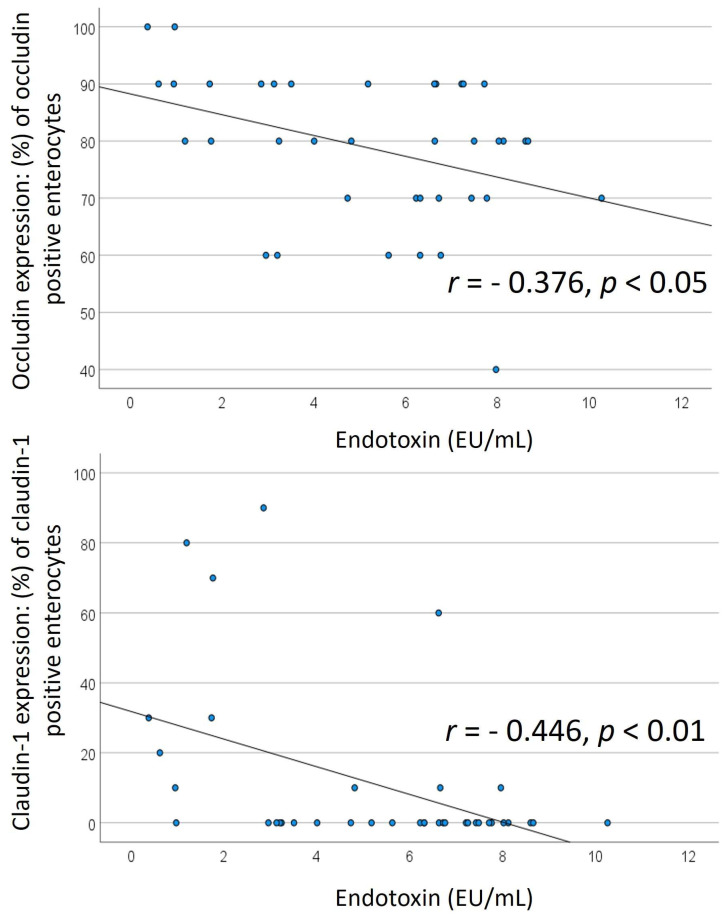
Correlation of tight junction proteins and endotoxin. Correlation of intestinal occludin and claudin–1 expression with blood endotoxin concentrations. The expression of occludin and claudin–1 in the intestinal mucosa was significantly inversely correlated with endotoxin concentration.

**Table 1 biomedicines-12-00160-t001:** Characteristics of healthy controls and patients with heart failure.

Characteristics	Controls (n = 10)	HFrEF (n = 14)	HFpEF (n = 14)	Statistics
Age	45.9 ± 16.3	65.21 ± 9.67 *	69.35 ± 10.42 **	*p* = 0.001
Gender	5 M/5 F	7 M/7 F	6 M/8 F	*p* = 0.913
Overweight (BMI 25–29.9)	3/10	3/14	4/14	*p* = 0.869
Smoking	2/10	3/14	2/14	*p* = 0.878
Hypertension	N/A	5/14	5/14	*p* = 1
Hypercholesterolemia	N/A	5/14	4/14	*p* = 0.685
Coronary artery disease	N/A	5/14	4/14	*p* = 0.685

The statistics are the result of the x^2^-test amongst the corresponding three groups, or two groups when the examined parameter is N/A to controls. * *p* < 0.05, ** *p* = 0.001 vs. controls. N/A, non-applicable.

**Table 2 biomedicines-12-00160-t002:** Histological features in the intestinal mucosa of healthy controls and heart failure patients.

Histopathological Features	Controls (n = 10)	HFrEF (n = 14)	HFpEF (n = 14)	Statistics
Apoptotic body count	0.1 (0.1–0.1)	0.1 (0.1–0.3)	0 (0–0.15)	*p* = 0.075
Mitotic count	0.5 (0.2–0.5)	0.4 (0.25–0.60)	0.4 (0.27–0.60)	*p* = 0.886
Villus length (mm)	0.39 ± 0.08	0.39 ± 0.09	0.40 ± 0.09	*p* = 0.904
Intraepithelial CD3+ lymphocytes/100 intestinal epithelial cells	15 (10–20)	15 (10–15)	10 (10–20)	*p* = 0.410

Values are median (IQR), except for villous length (mean ± SD).

**Table 3 biomedicines-12-00160-t003:** Gradient of occludin expression (% of occludin (+) enterocytes) along the crypt–villous axis in patients’ groups (values are mean ± SD).

Occludin Expression	Controls (n = 10)	HFrEF (n = 14)	HFpEF (n = 14)	Statistics
Part of the villi				
Τip	84 ± 9.66	48.57 ± 34.16 *	40.71 ± 30.50 **	*p* = 0.003
Μiddle	94 ± 6.99	82.14 ± 18.47	82.14 ± 8.02 **	*p* = 0.011
Crypt	97 ± 4.83	91.43 ± 8.64	92.14 ± 6.99	*p* = 0.142
Total expression	90 ± 6.67	75 ± 13.45 **	74.29 ± 11.58 **	*p* = 0.002

The statistics are the result of the Kruskal–Wallis test. The results of statistically significant pairwise comparisons are indicated by * *p* < 0.05, ** *p* < 0.01 vs. controls.

**Table 4 biomedicines-12-00160-t004:** Cytokine levels in peripheral blood.

Cytokine	Controls (n = 10)	HFrEF (n = 14)	HFpEF (n = 14)	Statistics
IL-1β(pg/mL)	0 (0–0.58)	0 (0–0)	0 (0–0)	*p* = 0.706
IL-6 (pg/mL)	0.57 (0.42–1.51)	6.28 (4.21–11.02) *	10.14 (9.33–24.44) **	*p* < 0.001
IL-8 (pg/mL)	3.3 (2.40–6.94)	24.25 (10.33–29.42) *	23.7 (15.80–116.25) *	*p* = 0.008
IL-10 (pg/mL)	0.24 (0.16–0.40)	1.28 (0.20–1.71) *	1.83 (1.28–2.80) **	*p* < 0.001
IL-12 (pg/mL)	0 (0–0)	0 (0–0)	0 (0–0)	*p* = 0.270
TNF-a (pg/mL)	0 (0–0.40)	0 (0–0)	0 (0–7.10)	*p* = 0.699

Values are median (IQR). The statistics are the result of the Kruskal–Wallis test. The results of statistically significant pairwise comparisons are indicated by * *p* < 0.05, ** *p* < 0.001 vs. controls.

## Data Availability

Data of this paper are available at Editor’s request.

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
