# Peer review of "Altered Expression of Intestinal Tight Junction Proteins in Heart Failure Patients with Reduced or Preserved Ejection Fraction: A Pathogenetic Mechanism of Intestinal Hyperpermeability"

_biomedicines, 2024, doi:10.3390/biomedicines12010160_

Round 1

Reviewer 1 Report

Comments and Suggestions for Authors

Koufou et al performed an original approach to one of the systemic complications of heart failure, the intestinal dysfunction being a hallmark of poor prognosis in these patients.

The manuscript is decently written, however I have some observations:

Based on which recommendations did you performed the endoscopy of the healthy subjects? Nevertheless, it is an invasive procedure with the subsequent risks. If ulcer or any other cause of digestive tract disease was suspected, this would come under exclusion criteria, and not “healthy at all”. Otherwise, please provide the study protocol and the patient consent form that was used in performing an endoscopy in research only purpose.

Please provide the exact statistic. NS (i.e. not significant difference is very vague, not significant may be both 0.061 and 0.934, but in the former the difference exists and is sizeable between the 2 study groups).

Table 4. I don’t understand the result 0 for several cytokines in both subgroups. Are you sure that the dilutions were adequately performed for the specified detection range? Please provide additional data in the materials and methods section.

It would be interesting to discuss the role of probiotic supplements in patients with HF and how could these molecules influence the clinical and biological profile. Moreover, a paragraph or two concerning the prophylaxis with rifaximin (such in the case of cirrhotic patients) should add certain interest to readers.

Best regards,

The Reviewer

Comments on the Quality of English Language

English is fine

Author Response

- Based on which recommendations did you performed the endoscopy of the healthy subjects? Nevertheless, it is an invasive procedure with the subsequent risks. If ulcer or any other cause of digestive tract disease was suspected, this would come under exclusion criteria, and not “healthy at all”. Otherwise, please provide the study protocol and the patient consent form that was used in performing an endoscopy in research only purpose.

Author response: We apologize for the non-clear description of the healthy control group. The control group consisted of people without anyone of the exclusion criteria as well as without known comorbidities, who underwent an upper gastrointestinal tract endoscopy due to symptoms of dyspepsia, after consultation of a gastroenterologist, without any pathological findings. Information regarding the control group was written clearer now in lines 99-102.

-Please provide the exact statistic. NS (i.e. not significant difference is very vague, not significant may be both 0.061 and 0.934, but in the former the difference exists and is sizeable between the 2 study groups).

Author response: Thank you for your right suggestion, which improves the clarity of presentation of our results. The exact P values have been provided now in an extra column (“statistics”) which has been added in all tables (1-4) and highlighted with yellow color. In table 1, an oversight regarding age has been corrected; although HFrEF and HFpEF presented no significant difference between them, controls were significantly younger (which is reasonable as controls were healthy persons without comorbidities e.g. hypertension). Explanations on the statistic method used has been provided, apart from the Statistical analysis section, in tables’ footnotes as well.

- Table 4. I don’t understand the result 0 for several cytokines in both subgroups. Are you sure that the dilutions were adequately performed for the specified detection range? Please provide additional data in the materials and methods section.

Author response: The cytokine concentrations were measured by a Cytometric Bead Array methodology that combines the principles of ELISA and Flow Cytometry to quantify soluble proteins. As such, protein concentrations that fall below the standard curve of the assay cannot be safely quantified. The samples that fell below range were measured without dilution to ensure the accuracy of the results and subsequently reported as "0". You can find our revision highlighted with yellow in lines 114-118 of the section Materials and Methods.

-It would be interesting to discuss the role of probiotic supplements in patients with HF and how could these molecules influence the clinical and biological profile. Moreover, a paragraph or two concerning the prophylaxis with rifaximin (such in the case of cirrhotic patients) should add certain interest to readers.

Author response: Thank you for this very important suggestion. A new paragraph on the role of gut microbiota modulation as a therapeutic intervention in HF patients, with a special emphasis on probiotics supplementation or rifaximin administration has been added (lines 352-375) and conclusion was slightly modified to include this information (lines 395-396).

Reviewer 2 Report

Comments and Suggestions for Authors

I’ve read with attention the paper of Koufou et al. that is potentially of interest. The background and aim of the study have been clearly defined. The methodology applied is overall correct, the results are reliable and adequately discussed. I’ve only some minor comments:

- The abstract should include some main numerical results

- It is not clear how the authors established the number of patients to be enrolled per group

- Line 338: "measured"

- The reference style is not the one of the journal and many reference are not complete

- I would suggest to mildly reduce the self-citation rate, eventually adding other recent references from Biomedicines.

Author Response

-The abstract should include some main numerical results

Author response: We were forced to cut the numerical data from the abstract due to word number limitation, but now at your urging we have added them. You will find them highlighted with yellow color in lines 37-42.

- It is not clear how the authors established the number of patients to be enrolled per group

Author response: A power analysis to predefine the number of patients to be enrolled in each group was not performed because: (a) This was a novel, pilot, study, with several parameters under investigation. Our scope was to obtain preliminary data that can be used for planning definitive studies on this research area and (b) we would not be able to increase in a pilot study the number of patients enrolled due to the invasive procedures applied for tissue sampling (upper gastrointestinal tract endoscopy). In the Study Design section we have added now the information that our study is pilot (line 79).

- Line 338: "measured"

Author response: corrected (line 379 of the revised manuscript)

- The reference style is not the one of the journal and many reference are not complete

Author response: We have revised all the references’ style to comply with the journal’s requirements.

- I would suggest to mildly reduce the self-citation rate, eventually adding other recent references from Biomedicines.

Author response: According to your suggestion, we have changed some of our self-citations with two more adequate and recent citations of Biomedicines (reference 27 and 31 of the revised manuscript).

Round 2

Reviewer 1 Report

Comments and Suggestions for Authors

The authors addressed my previous observations.

Comments on the Quality of English Language

English seems fine.